# Use of Indocyanine Green (ICG), a Medical Near Infrared Dye, for Enhanced Fluorescent Imaging—Comparison of Organic Anion Transporting Polypeptide 1B3 (OATP1B3) and Sodium-Taurocholate Cotransporting Polypeptide (NTCP) Reporter Genes

**DOI:** 10.3390/molecules24122295

**Published:** 2019-06-21

**Authors:** Menq-Rong Wu, Yi-You Huang, Jong-Kai Hsiao

**Affiliations:** 1Institute of Biomedical Engineering, National Taiwan University, No. 1, Sec. 4, Roosevelt Rd., Taipei 10617, Taiwan; nicole750707@gmail.com (M.-R.W.); yyhuang@ntu.edu.tw (Y.-Y.H.); 2Department of Medical Imaging, Taipei TzuChi General Hospital, Buddhist Tzu-Chi Medical Foundation, No.289, Jianguo Rd., Xindian Dist., New Taipei city 23142, Taiwan; 3School of Medicine, Tzu Chi University, No. 701, Sec. 3, Zhongyang Rd. Hualien 97004, Taiwan

**Keywords:** sodium taurocholate cotransporting polypeptide (*NTCP*), organic-anion-transporting polypeptide 1B3 (*OATP1B3*), indocyanine green (ICG), in vivo imaging system (IVIS)

## Abstract

Molecular and cellular imaging in living organisms have ushered in an era of comprehensive understanding of intracellular and intercellular events. Currently, more efforts have been focused on the infrared fluorescent dyes that facilitate deeper tissue visualization. Both sodium taurocholate cotransporting polypeptide (NTCP) and organic-anion-transporting polypeptide 1B3 (OATP1B3) are capable of carrying indocyanine green (ICG) into the cytoplasm. We compared the feasibility of *NTCP* and *OATP1B3* as reporter genes in combination with ICG. *NTCP* and *OATP1B3* were transduced into HT-29 cells. Genetically modified HT-29 cells were inoculated into nude mice. ICG was administered in vitro and in vivo and the signals were observed under confocal microscopy, flow cytometry, multimode microplate reader, and an in vivo imaging system. Both *NTCP*- and *OATP1B3*-expressing cells and xenografts had higher ICG intensities. The OATP1B3-expressing xenograft has a higher ICG uptake than the *NTCP*-expressing xenograft. *NTCP* or *OATP1B3* combined with ICG could serve as a noninvasive imaging modality for molecular and cellular imaging. *OATP1B3* outperforms *NTCP* in terms of in vivo imaging.

## 1. Introduction

Molecular and cellular imaging in living organisms has ushered in an era of comprehensive understanding of intracellular and intercellular events. Labeling fluorescent tags or nanoparticles that exhibit magnetic resonance (MR)/computed tomography (CT)/positron emission tomography (PET) signals is convenient for short-term cell tracking. However, a dilutional effect exists because of cell division and metabolism [1,2]. Therefore, genetic modification with reporter genes is a superior approach for long-term repeated cell tracking. Currently, numerous reporter genes are applied in molecular imaging activities, such as traditional green fluorescent protein (GFP), red fluorescent protein (RFP), and bioluminescence imaging (BLI) [3,4]. Examples of reporter genes include herpes simplex virus type 1 thymidine kinase reporter and dopamine transporter in PET [5,6]; and transferrin receptor, organic-anion-transporting polypeptide 1B1 (*OATP1B1*), and organic-anion-transporting polypeptide 1B3 (*OATP1B3*) in magnetic resonance imaging (MRI) [7,8,9]. Fluorescence and BLI are advantageous based on their sensitivity compared with other imaging modalities. However, the penetration path depth is limited due to the nature of optics. Currently, more efforts are being focused on infrared fluorescent dyes that facilitate much deeper tissue visualization. Indocyanine green (ICG), one of the best candidates, has been used extensively in clinical practice [10,11,12,13]. The excitation/emission wavelength of ICG is 760/830 nm, facilitating superior tissue penetration compared with conventional near-infrared (NIRF) dyes such as Cy5.5 (excitation/emission wavelength of 675 nm/694 nm) [14,15,16].

Both sodium taurocholate cotransporting polypeptide (NTCP) and OATP1B3 are capable of carrying ICG into the cytoplasm [17]. OATP1B3 belongs to the solute carrier organic anion transporter family (OATP family) and is expressed in hepatocytes. It is a transmembrane glycoprotein responsible for transporting bilirubin, nutrients, drugs, specific types of MRI agents such as gadolinium-ethoxybenzyl-diethylenetriaminepentaacetic acid (Gd-EOB-DTPA), and ICG [17,18]. NTCP, which exhibits similarity with OATP1B3, is a sodium-dependent transporter that belongs to the solute carrier family of transporters (SLC10). It transports bile salts, sulfated compounds, thyroid hormones, drugs, toxins, and Gd-EOB-DTPA [18,19]. NTCP and OATPs are responsible for the transfer of bile salts from plasma into the liver [20]. Moreover, recent studies have reported a novel characteristic of NTCP as an entry transporter of hepatitis B virus (HBV) and hepatitis D virus (HDV) [21].

We have previously reported the efficacy of OATP1B3 combined with ICG as a reporter gene/fluorescent source imaging system that is capable of imaging OATP1B3 activity in a HT-1080 tumor-bearing nude mouse for up to 96 h, a long period compared to other methods. ICG has a fast-binding property with plasma proteins, enters cells through transporters such as OATP1B3 and NTCP, and is excreted by the liver into bile juice rapidly [17,22]. We have also investigated NTCP as a reporter for monitoring the treatment of HBV infections [23]. However, the feasibility of manipulating *NTCP* as a reporter is yet to be compared with *OATP1B3* in combination with the ICG imaging system. The goal of the study was to compare the reporter system combined with ICG in *NTCP* and *OATP1B3*.

## 2. Results

### 2.1. Validation of NTCP/OATP1B3 Overexpression

A HT-29 cell line was infected with *NTCP*- or *OATP1B3*-expressing lentivirus to generate NTCP- or OATP1B3-expressing HT-29 cells. To determine the transduction efficiency, we evaluated the protein levels of NCTP or OATP1B3 in the cells using Western blotting. *NTCP*- or *OATP1B3*-overexpressing HT-29 cells carried more NTCP or OATP1B3 (Appendix A). The NTCP and OATP1B3 in the cells were colocalized with phalloidin, indicating that they were mainly in the cellular membrane (Appendix A). Moreover, NTCP was also expressed in the cytoplasm and the nucleus (Appendix A). In addition, cell viability did not change following transduction (Appendix A).

### 2.2. Evaluation of ICG Transporting Capacity in NTCP and OATP1B3 In Vitro

After treatment of the control, and *NTCP*- and *OATP1B3*-expressing HT-29 cells with ICG, *NTCP*-expressing HT-29 cells had a stronger ICG signal than *OATP1B3*-expressing HT-29 cells (Figure 1). The intensity ratio of ICG observed under flow cytometry was approximately 1:2:5 in the control, *OATP1B3*-, and *NTCP*-expressing HT-29 cells (Figure 1a). To assess the detection limits of the in vivo imaging system (IVIS), we determined series cell numbers following ICG treatment. The minimum cells that could be detected were 3.13 × 10^3^ and 1.25 × 10^4^ in *NTCP*- and *OATP1B3*-expressing HT-29 cells, respectively (Figure 1b). In addition, the ICG incubation time was positively related to the ICG intake capacity in *NTCP*- and *OATP1B3*-expressing HT-29 cells (Figure 2a). To observe the retention capacity of ICG in the control and in the *NTCP*- and *OATP1B3*-expressing HT-29 cells, we detected the ICG signals at different times after ICG treatment. In the beginning, the ICG intensity in *NTCP*-expressing HT-29 cells was approximately two-fold the ICG intensity of *OATP1B3*-expressing HT-29 cells. With time, the ICG intensity in *NTCP*-expressing HT-29 cells dropped to levels lower than in *OATP1B3*-expressing HT-29 cells. Therefore, 72 h after ICG treatment, ICG intensity in the *OATP1B3*-expressing HT-29 cells was two times greater than the ICG intensity in *NCTP*-expressing HT-29 cells (Figure 2b).

### 2.3. Evaluation of ICG Transporting Ability between NTCP and OATP1B3 In Vivo

After nude mice were administrated with the control and *NTCP*/*OATP1B3*-expressing HT-29 cells on each side near the hind limb, we administered ICG intraperitoneally to track the ICG signal in the following days. Both *NTCP*- and *OATP1B3*-expressing tumor cells had higher ICG signals than the control tumor cells. In addition, the *OATP1B3*-expressing tumor cells had higher ICG signals than the *NTCP*-expressing tumor cells (Figure 3a).

The *OATP1B3*-expressing tumor cells had the highest ICG intensity followed by the *NTCP*-expressing tumor cells and the control tumor cells (Figure 3b). To obtain an enhanced intensity ratio in *NTCP*- and *OATP1B3*-expressing tumor cells, we compared their normalized ICG intensities with the ICG intensity of control tumor cells. After normalization was completed, ICG intensity in the *NTCP*-expressing tumor cells was approximately 1.5 times that of the control tumor cells at all time points. In addition, ICG intensity in *OATP1B3*-expressing tumor cells was two times higher at 4 h and approximately 4 times higher at 24, 48, and 72 h compared to the ICG intensity of control tumor cells (Figure 3c).

### 2.4. Assessment of Gd-EOB-DTPA Intake In Vivo in MRI

OATP1B3 and NTCP are both Gd-EOB-DTPA transporters [18]; therefore, they could be considered dual reporter proteins. However, MRI revealed that *NCTP*-expressing cells did not exhibit the Gd-EOB-DTPA signal in vitro (Figure 4a). In the IVIS modality, both *NTCP*- and *OATP1B3*-expressing tumor cells had slightly elevated ICG signals (Figure 4c). In addition, *OATP1B3*-expressing tumor cells had higher MR imaging signals (Figure 4c). The MR signal intensity was approximately two times the intensity in *OATP1B3*-expressing tumor cells compared with the control and *NTCP*-expressing tumor cells (Figure 4b). Moreover, tumor size was not a factor influencing MRI signal intensity (Figure 4d).

### 2.5. Evaluation of ICG Transporting Capacity between NTCP and OATP1B3 In Vivo

At 2 d after ICG administration, the *NTCP*- or *OATP1B3*-expressing tumor-bearing mice were sacrificed to observe the biodistribution of ICG through IVIS. In the *NTCP*- and *OATP1B3*-expressing tumor-bearing mice, *OATP1B3*-expressing tumor cells contained considerable amounts of ICG, followed by the liver, *NTCP*-expressing tumor, and the control tumor. Other organs had minimal amounts of ICG (Figure 5a,b). In addition, NTCP and OATP1B3 proteins were mainly expressed in *NTCP*- and *OATP1B3*-expressing tumors, respectively (Figure 5c,d).

## 3. Discussion

The experiments described in the present study illustrate novel reporter genes that present a novel imaging modality. Both *NTCP* and *OATP1B3* combined with ICG could be applied in near infrared (NIR) imaging modalities such as IVIS and photoacoustic imaging. In addition, OATP1B3 could serve as an MR reporter gene for MRI applications.

Our results showed that *NTCP* and *OATP1B3* are safe reporter genes because *NTCP* and *OATP1B3* transduction did not influence cell viability and tumor size (Appendix A). In addition, ICG is a Food and Drug Administration (FDA)-approved NIR compound in which safety margins have been explored extensively. It maintains cell viability, which minimizes potential side effects in the course of drug–drug interaction experiments. The characteristics of NIR and their accessibility in laboratories make cell monitoring using the techniques presented herein achievable.

The single modality of imaging the acquired discrete information limits our capacity to examine tumors comprehensively. Through IVIS combined with MRI, we could obtain more information about tumor growth and tumor status. In addition, we could perform experiments within the minimal animals using noninvasive imaging modalities over relatively short time frames. In our reporter system, we could assess cells using noninvasive imaging modalities at different times in a single animal, which minimized individual error and resources in terms of experiment time and experimental animal costs.

The ICG intake capacity was superior in *NTCP*-expressing cells to *OATP1B3*-expressing cells at the beginning, indicating that NTCP had superior ICG transporting capacity. However, with time, *NTCP*-expressing cells contained smaller amounts of ICG than *OATP1B3*-expressing cells (Figure 2b). The efflux of ICG is mediated by a multidrug resistance p-glycoprotein (MDR)-3 [24]. No evidence is available to confirm that the overexpression of *NTCP* enhances the expression of *MDR3*. Therefore, we deduced that it could be due to the disorientation of the NTCP on the cell membrane. Under normal status, the orientation of NTCP is outside-in. In *NTCP*-expressing cells, some NTCP could be oriented inside-out so that ICG is pumped out. Therefore, *NTCP*-expressing cells had lower ICG retention capacity compared with *OATP1B3*-expressing cells. It has been reported that OATP1B3 and NTCP are Gd-EOB-DTPA transporters (for OATP1B3 [K_m_ = 4.1 mM, V_max_ = 22.7 pmol/mg · min], and for NTCP [K_m_ = 0.04 mM, V_max_ = 1.4 pmol/mg · min]) [18]; however, the ingested Gd-EOB-DTPA was in trace amounts that could not be detected using clinical 1.5T MRI (Figure 4). The results demonstrated the varied transportation mechanisms between NTCP and OATP1B3. In addition, *OATP1B3*-transduced cells could integrate both ICG and EOB-Gd-DTPA, which is ideal for combined IVIS and MRI systems. Therefore, *OATP1B3* is a superior reporter gene for cell tracking and tumor monitoring.

At 4 h after ICG was injected into *NTCP*/*OATP1B3*-expressing tumor-bearing mice, the ICG signal was detected in the whole body and accumulated in the *NTCP*/*OATP1B3*-expressing tumors at 24 h after injection (Figure 3a). The background uptake in most tissues was presumed to be low because *NTCP* and *OATP1B3* exhibit restricted tissue expression [25,26].

Sodium fluorescein (NaFluo) and numerous fluorescein derivatives, which have potential application in in vivo imaging probes, have been identified as substrates mediated by OATP1B1 and OATP1B3 [27]. Considering the structural similarity between NaFluo and fluorescein isothiocyanate (FITC), we tested whether FITC could be applied as an in vivo tracking probe. *OATP1B3*-expressing HT-29 could assimilate more FITC, as revealed using flow cytometry (Appendix A); however, it may not have the ability to be applied in in vivo imaging (Appendix A) owing to the nonspecific binding of FITC to the entire tissue. Based on the results of ICG and FITC application as probes in the *OATP1B3* reporter system, ICG was a superior probe for in vivo imaging (Figure 3a and Appendix A).

## 4. Materials and Methods

### 4.1. Cell Line and Culture

HEK 293T and HT-29 cells were cultured in Dulbecco’s modified Eagle medium (Thermo Fisher Scientific, Waltham, MA, USA). We added 10% fetal bovine serum (Biologic Industries, Cromwell, CT, USA), 100 U/mL penicillin, and 100 mg/mL streptomycin (Thermo Fisher Scientific). These cells were cultured in a humidified atmosphere incubator containing 5% CO_2_ at 37 °C.

### 4.2. Vector Construction and Cell Transfection and Transduction

Virus production was performed according to the method in Nature Protocols [28]. The plasmids were pWPXL-OATP1B3-ires-Puro and pWPXL-NTCP-ires-Puro that we constructed previously [23]. HT-29 cells (2 × 10^5^) were seeded in 6-well plates for 1 d, which were transduced at a multiplicity of infection of 5. Subsequently, the cells were selected with 2 μg/mL puromycin (Millipore-Sigma, Billerica, MA, USA) for 10 d at 1 d after infection.

### 4.3. Evaluations on the Cellular Uptake of ICG

To test the detecting limitation of the IVIS Spectrum imaging system, 2 × 10^6^ cells were treated with 50 µg/mL ICG (Millipore-Sigma, Billerica, MA, USA) for 1 h after the spray in 6-well plates for 1 day. The excess ICG was then washed out 3 times with 1× phosphate-buffered saline (PBS). There were series diluted cells in 96-well black plates for detecting the ICG signal via an IVIS Spectrum imaging system (Xenogen, Perkin Elmer, Waltham, MA, USA). All the data were acquired under similar parameters (emission/excitation channel: ICG/ICG; exposure time: 1 min; binning: medium; lens aperture [f/stop]: 2; field of view: 12 cm). The images were presented using radiant efficiency units (p/s/cm^2^/sr)/(µW/cm^2^). The quantification of ICG intensity was based on the average efficiency in the region of interest (ROI).

For intake capacity, 1 × 10^6^ cells were treated with 50 µg/mL ICG for 1 h after the spray in 12-well plates for 1 day. Cells were trypsinized, neutralized, and then washed 3 times with PBS. The ICG signal was acquired using a FACSCalibur (BD Biosciences, San Jose, CA, USA). The filter used for the detection was an APC-Cy7 channel (785 nm) filter.

To compare the ICG uptake ability, 2 × 10^4^ cells were seeded in 96-well plates for 1 day before treatment with 50 µg/mL ICG for 1, 5, 10, 30, and 60 min. After the excess ICG was washed out 3 times with 1× PBS, cells were detected using Spark 10M (Tecan Trading AG, Männedorf, Switzerland) at 0, 1, 3, 6, 24, 48, and 72 h after ICG treatment. In addition, the ICG signal was observed under a TCS SP5 laser-scanning microscope (Leica, Wetzlar, Germany) with a Cy5 filter.

### 4.4. Animal Experiments

We purchased 6 to 8-week-old female BALB/cAnN.Cg-Foxnlnu/CrlNarl nude mice from the National Laboratory Animal Center. The experimental study was approved by the Institutional Animal Use and Care Committee of Taipei Tzu Chi Hospital, Buddhist Tzu Chi Medical Foundation (106-IACUC-004). The maintenance of the mice followed the suggestions of the Guide for the Care and Use of Laboratory Animals (National Institues of Health, NIH).

### 4.5. Xenograft

HT-29 control cells and *NTCP*-expressing cells (2.5 × 10^6^) were injected subcutaneously to the nude mice on the left and right side, respectively. The cell number used to inoculate control cells and *OATP1B3*-expressing cells was 1.0 × 10^6^. We measured the tumor sizes using a digital caliper (Shineteh Instruments, Taipei, Taiwan) before fluorescence imaging and MRI were conducted and estimated using the following formula: [(longest length) × (shortest)^2^]/2.

### 4.6. Fluorescence and Bioluminescence Imaging In Vivo and Ex Vivo

Tumor-bearing nude mice were administrated 10 mg/kg ICG (dissolved in ddH_2_O) intraperitoneally and observed using in vivo fluorescence imaging at 4, 24, 48, and 72 h after injection using an IVIS Spectrum imaging system (Xenogen; Perkin Elmer). The mice were sacrificed to examine the ICG signal intensity in tumors and organs under IVIS after 10 mg/kg ICG was injected for 2 d. We acquired all images using the same parameters [emission/excitation channel: ICG/ICG; exposure time: auto; binning: medium; lens aperture: 2; field of view: 12 cm]. The imaging data acquired from IVIS are quantified in units of radiant efficiency, which means photons per second per square cm per steradian/microwatts per square cm. The quantification of ICG intensity was based on the average efficiency in the ROI.

### 4.7. MRI In Vitro

Cells (5 × 10^5^) were seeded in 6-well plates overnight, which were treated with 1.25 mM Gd-EOB-DTPA (Bayer Pharma AG, Berlin, Germany) for 4 h. Cells were PBS washed 3 times using trypsinized. After cells were centrifuged at 1200 rpm for 5 min in 0.2-mL tubes at 4 °C, the cells were analyzed using a clinical 1.5T MRI system (Siemens Magnetom Aera, Erlangen Germany). We used 2-dimensional T1-weighted fast spin-echo pulse sequences (TR/TE = 700/20 ms). The scanning slice thickness was 1.0 mm with a 0.5-mm gap, and the field of view (FOV) was 14 cm × 6 cm. Moreover, the matrix size was 256 × 112.

### 4.8. MRI In Vivo

Nude mice underwent xenografting for 11 days. Images were acquired 1 h after 200 μL of 250 mM Gd-EOB-DTPA was administered intravenously. Images were acquired using a 7T-MRI system (Bruker Biospec 70/30, Ettlingen, Germany). Fast spin-echo pulse sequences (FSE) provided by the vendor were used (TR/TE = 842/8.6404 ms; resolution = 256 × 256). The slice thickness was 0.5 mm. The FOV was 5 cm × 5 cm. The total scanning time was 3 min and 20 s (Number of excitation (NEX) = 10). The images were analyzed using RadiAnt DICOM Viewer (64-bit) (Medixant, Poznan, Poland).

### 4.9. Western Blot Analysis

Immunoblotting analysis was performed as described previously [9]. Primary antibodies were against NTCP (1:1000; Thermo Fisher Scientific), α-tubulin (1:5000; Merck Millipore, Burlington, MA, USA), OATP8 (1:1000; Santa Cruz Biotechnology, Dallas, TX, USA), and Glyceraldehyde-3-Phosphate Dehydrogenase (GAPDH) (1:5000; Cell Signaling Technology, Danvers, MA, USA) separately at 4 °C and were then incubated with 1:5000 horseradish peroxidase-conjugated rabbit/mouse anti-IgG for 1 h at room temperature.

### 4.10. Immunofluorescence

After the cells were fixed in 4% formaldehyde, they were blocked with 5% BSA. Next, the cells were incubated with 1:100 rabbit polyclonal anti-OATP8 antibodies (Santa Cruz, CA, US) and 1:100 anti-NTCP antibodies (Thermo Fisher Scientific, Waltham, MA, USA) at 4 °C overnight. After 4 times washing in PBST, the slides were treated with 488-conjugated goat anti-rabbit IgG antibodies (Thermo Fisher Scientific, Waltham, MA, USA), DAPI, and rhodamine phalloidin (Thermo Fisher Scientific, Waltham, MA, USA) at room temperature for 1 h. All cover slides were visualized using a Fluorescent Cell Imager (ZOE, Bio-Rad, Hercules, CA, USA).

### 4.11. Immunohistochemistry

Immunohistochemical analysis was performed as described previously [9]. The antibodies were anti-OATP8 (1:100 in 1% BSA; Santa Cruz, CA, USA) and anti-NTCP (1:1000; Thermo Fisher Scientific) antibodies and analysis was performed at 4 °C overnight. All slides were examined under an ECLIPSE TE2000-U microscope (Nikon, New York, NY, USA).

### 4.12. Statistical Analyses

All data were illustrated as the means ± standard error of the mean (SEM) from at least 3 independent experiments. The Newman—Keuls Test and Dunnett’s Multiple Comparison Test were used to determine significance among differences; * *p* < 0.05; ^#^
*p* < 0.01; and ^&^
*p* < 0.001.

## 5. Conclusions

We demonstrated the potential application of NTCP or OATP1B3 and ICG as novel combinations for noninvasive imaging modalities. In addition, OATP1B3 could serve as a dual reporter gene for MR imaging and optical fluorescence imaging.

## Figures and Tables

**Figure 1 molecules-24-02295-f001:**
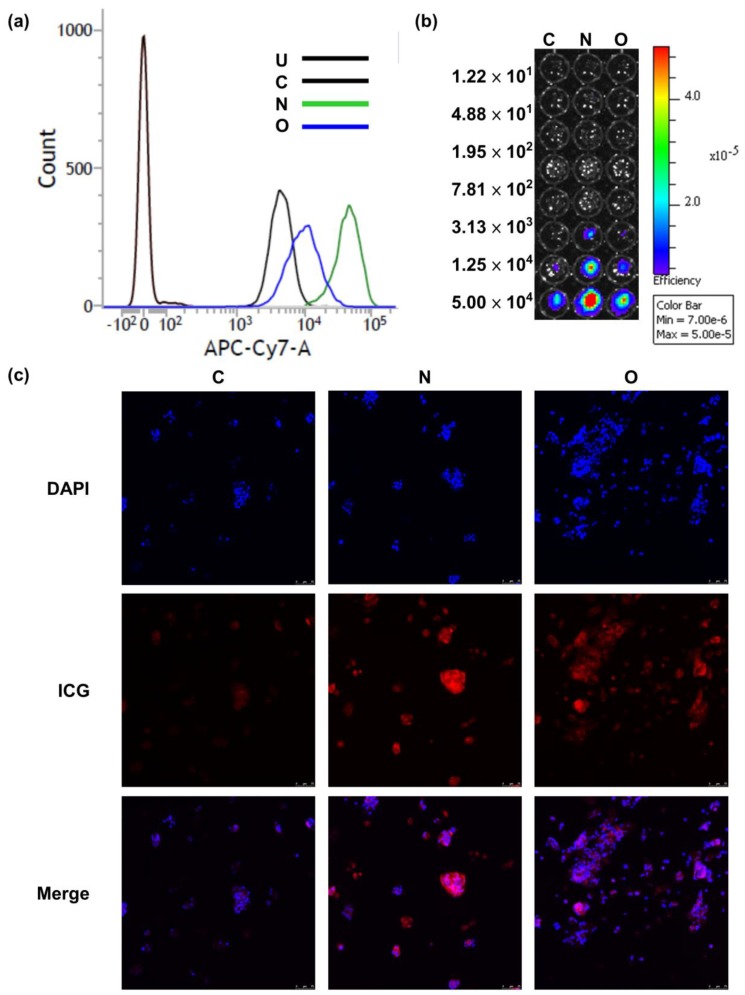
**I**ndocyanine green (ICG) intake capacity in sodium taurocholate cotransporting polypeptide (*NTCP*)-expressing and organic-anion-transporting polypeptide 1B3 (*OATP1B3*)-expressing HT-29 cells. (**a**) The cells were measured by using flow cytometry at the APC-Cy7 wavelength (785 nm) to observe ICG intensity. (**b**) The diluted series cells were detected using an in vivo imaging system (IVIS). (**c**) One hour after ICG was added to the control and the *NTCP*- and *OATP1B3*-expressing HT-29 cells, the cells were monitored under a confocal microscope. C: control HT-29. U: untreated. N: *NTCP*-expressing HT-29 cells. O: *OATP1B3*-expressing HT-29 cells.

**Figure 2 molecules-24-02295-f002:**
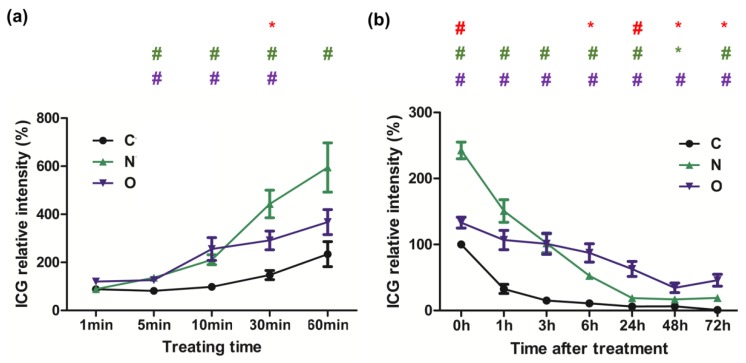
ICG intake and retention in *NTCP*- and *OATP1B3*-expressing HT-29 cells. (**a**) ICG intensity was detected after ICG treatment for 1, 5, 10, 30, and 60 min in the control and the *NTCP*- and *OATP1B3*-expressing HT-29 cells. N = 3. (**b**) ICG intensity was detected at 0, 1, 3, 6, 24, 48, and 72 h after ICG treatment for 1 h in the control and *NTCP*- and *OATP1B3*-expressing HT-29 cells. N = 3. C: control HT-29 cells. N: *NTCP*-expressing HT-29 cells. O: *OATP1B3*-expressing HT-29 cells. Error bars represent the standard error of the mean (SEM). * *p* < 0.05, ^#^
*p* < 0.01. * in purple color indicates control compared with *OATP1B3*-expressing HT-29 cells. * and ^#^ in green color indicate control compared with *NTCP*-expressing HT-29 cells. * and ^#^ in red color indicate *NTCP*-expressing HT-29 cells compared with *OATP1B3*-expressing HT-29 cells.

**Figure 3 molecules-24-02295-f003:**
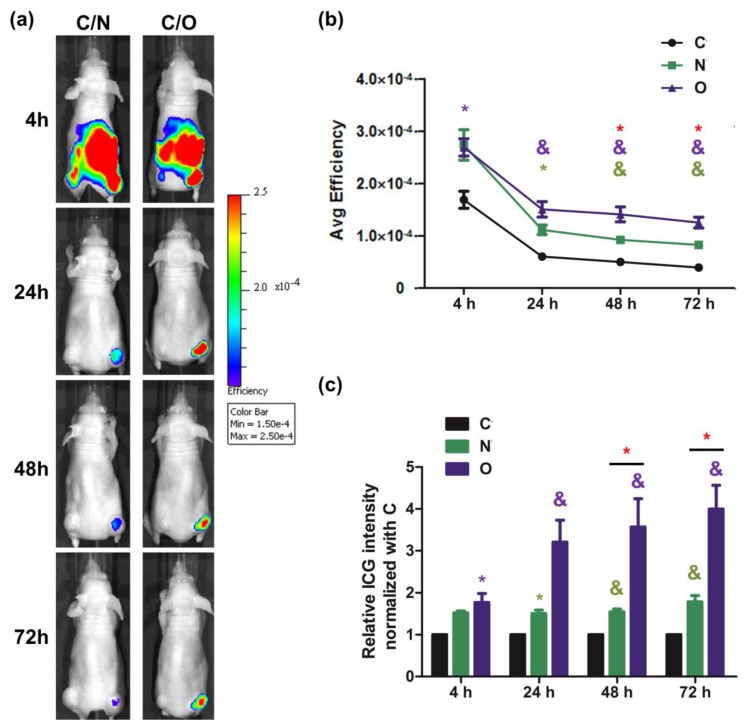
Evaluation of ICG intake in vivo. (**a**) The ICG signal was observed using IVIS at 4, 24, 48, and 72 h after injecting ICG to *NTCP*- and *OATP1B3*-expressing tumor-bearing mice. The tumor on the left hind leg was a controlled tumor, and on the right was an *NTCP*-expressing tumor at the top and an *OATP1B3*-expressing tumor at the bottom. (**b**) The relationship between tumor size and the total efficiency of ICG at 48 h after injecting ICG to *NTCP*- and *OATP1B3*-expressing tumor-bearing mice. (**c**) After the ICG signal was normalized based on control tumor size to obtain relative ICG intensity, we compared the relative ICG intensities in the control and *NTCP*- and *OATP1B3*-expressing tumor cells at 4, 24, 48, and 72 h after injecting ICG into the *NTCP*- and *OATP1B3*-expressing tumor-bearing mice. N = 3 and N = 4 for NTCP and OATP1B3 tumor-bearing mice, respectively. C: control tumor. N: *NTCP*-expressing tumor. O: *OATP1B3*-expressing tumor. Error bars indicate the SEM. * *p* < 0.05, ^#^
*p* < 0.01, ^&^
*p* < 0.001. *, ^#^, ^&^ in purple and green color indicate the control compared with *OATP1B3*- and *NTCP*-expressing tumor cells, respectively. * in red color indicates the *NTCP*-expressing tumor cells compared with *OATP1B3*-expressing tumor cells. Ctrl: control tumor cells.

**Figure 4 molecules-24-02295-f004:**
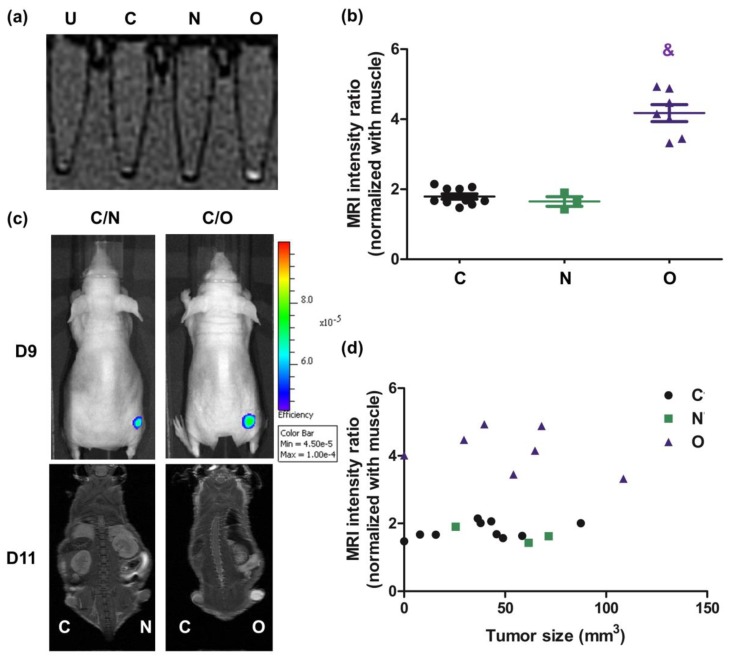
MRI contrast-Gd-EOB-DTPA intake in vivo in magnetic resonance imaging (MRI). (**a**) After HT-29 cells were transduced into *NTCP* and *OAT1B3* independently, treated with Gd-EOB-DTPA (MRI contrast) for 4 h, and detected using 3T-MRI scanning. (**b**) The relative signal intensity of the control and *NTCP*- and *OATP1B3*-expressing xenografts adjusted based on the signal intensity of muscles in MR images. Error bars indicate the SEM. N = 8, N = 3, and N = 5 in control, *NTCP*-expressing tumor cells, and OATP1B3-expressing tumor cells, respectively. (**c**) ICG intensity was acquired using IVIS 9 d after xenograft. MRI intensity was detected using 7T-MRI approximately 1 h after injecting MRI contrast into *NTCP*- and *OATP1B3*-expressing tumor-bearing mice 11 d after xenografting. The tumor on the left hind leg was the control tumor, and on the right was the *NTCP*-expressing tumor (at the top) and the *OATP1B3*-expressing tumor (at the bottom). (**d**) The relationship between tumor size and MRI signal at 1 h after ICG injection into *NTCP*- and *OATP1B3*-expressing tumor-bearing mice. C: control HT-29 cells or tumor. N: *NTCP*-expressing HT-29 cells or tumor. O: *OATP1B3*-expressing HT-29 cells or tumor. ^&^
*p* < 0.001.

**Figure 5 molecules-24-02295-f005:**
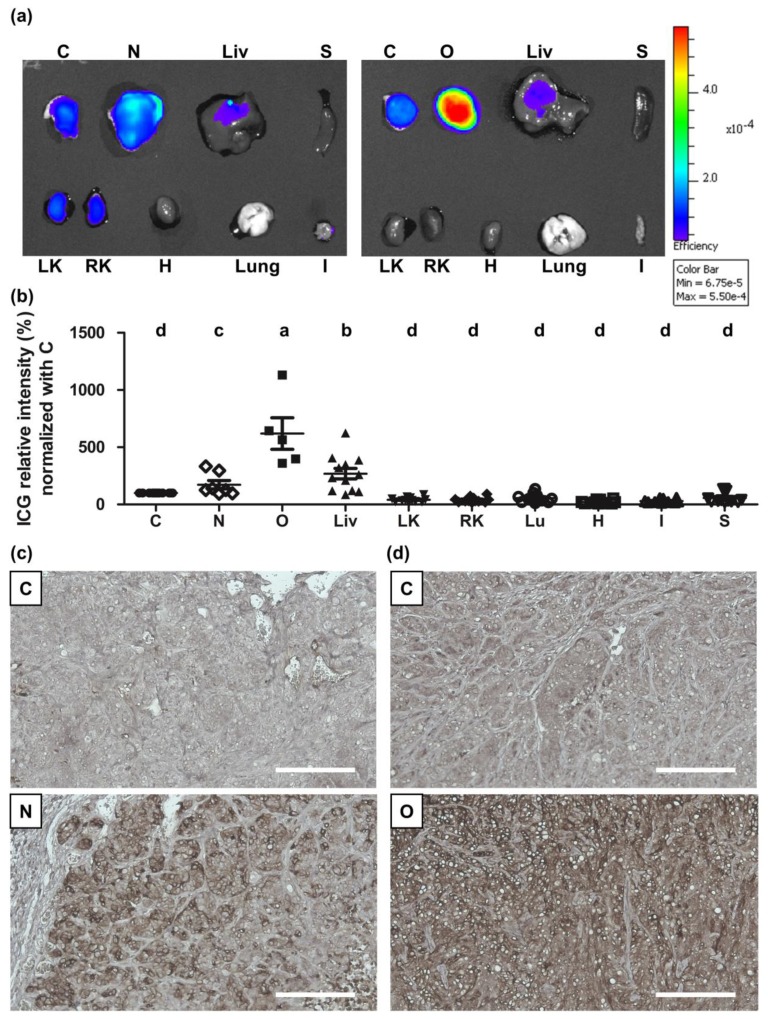
ICG intake ex vivo in the IVIS. After the administration of ICG for 2 d, we sacrificed the mice to evaluate the ICG signals using IVIS. (**a**) Ex vivo IVIS of *NTCP*- and *OATP1B3*-expressing tumor-bearing mice. C: control tumor; N: *NTCP*-expressing tumor; O: *OATP1B3*-expressing tumor. Liv: liver; LK: left kidney; RK: right kidney; Lu: Lung; H: heart; S: spleen; I: small intestine. (**b**) Quantification of ex vivo IVIS data from *NTCP*- and *OATP1B3*-expressing tumor-bearing mice; N = 7 and N = 5, respectively. Error bars indicate the SEM. a, b, c, and d indicate statistically significant differences among groups. (**c**) Tumors stripped from control and *NTCP*-expressing tumor-bearing mice stained for NTCP. (**d**) Tumors stripped from control and *OATP1B3*-expressing tumor-bearing mice stained for NTCP. The nuclei were stained using hematoxylin. Scale bar: 100 μm.

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
