# Peer review of "Use of Indocyanine Green (ICG), a Medical Near Infrared Dye, for Enhanced Fluorescent Imaging—Comparison of Organic Anion Transporting Polypeptide 1B3 (OATP1B3) and Sodium-Taurocholate Cotransporting Polypeptide (NTCP) Reporter Genes"

_molecules, 2019, doi:10.3390/molecules24122295_

Round 1

Reviewer 1 Report

The manuscript entitled "Use of indocyanine green (ICG), a medical near infrared dye, for enhanced fluorescent imaging organic anion transporting polypeptide 1B3 (OATP1B3) and sodium-taurocholate cotransporting polypeptide (NTCP) reporter genes" by Wu et.al, was quite interesting work. In this research, authors used two reporter genes of NTCP- and OATP1B3- that expressed on HT-29 cells and xenografts to measure and compare the intensities of NIR signal after administration of the ICG. From this study, authors concluded, both NTCP/ICG and OATP1B3/ICG combination could serve as a better noninvasive imaging modality for molecular and cellular imaging. Although, OATP1B3-expressing xenograft has a higher ICG uptake than the NTCP-expressing xenograft.

Overall, this study could be useful for molecular imaging research in pre-clinical set-up, especially ICG based dye has been already approved by the FDA, and so it may have more potential for the future clinical translation as well as dual modality imaging. Based on these merits, I would like to recommend this manuscript can be accepted for the publication in Molecules, after addressing the following minor modifications commented below.

Comments:   

01) Please use the same acronym of all the figures and legends in the manuscript for e.g., Control=C in figure 1, on the other hand in figure 5, it was mentioned as "CT" similarly experiment groups for "N" and "O" was replaced with NT and OT. Similarly in figure 4 legend, indicated as Ctl/C, please clarify it.  This need to be fixed. 

02) Figure 3C 'y" axis indicated as normalized with CT = control..?,

03) Also in the same legend, "SEM. *p <0.05, # p <0.01, p <0.001" symbol for p <0.001= "&" need to include.     

04) In Figure S1 legend, Western blot analysis NTCP (38kDa) and internal control a-tubulin (50kDa) in control and NTCP expressing HT-29. But in image it shows GAPDH as an internal control, which one is correct?   

Author Response

The manuscript entitled "Use of indocyanine green (ICG), a medical near infrared dye, for enhanced fluorescent imaging organic anion transporting polypeptide 1B3 (OATP1B3) and sodium-taurocholate cotransporting polypeptide (NTCP) reporter genes" by Wu et.al, was quite interesting work. In this research, authors used two reporter genes of NTCP- and OATP1B3- that expressed on HT-29 cells and xenografts to measure and compare the intensities of NIR signal after administration of the ICG. From this study, authors concluded, both NTCP/ICG and OATP1B3/ICG combination could serve as a better noninvasive imaging modality for molecular and cellular imaging. Although, OATP1B3-expressing xenograft has a higher ICG uptake than the NTCP-expressing xenograft.

Overall, this study could be useful for molecular imaging research in pre-clinical set-up, especially ICG based dye has been already approved by the FDA, and so it may have more potential for the future clinical translation as well as dual modality imaging. Based on these merits, I would like to recommend this manuscript can be accepted for the publication in Molecules, after addressing the following minor modifications commented below.

Ans: We deeply appreciate your comments and suggestions.

Comments:   

01)  Please use the same acronym of all the figures and legends in the manuscript for e.g., Control=C in figure 1, on the other hand in figure 5, it was mentioned as "CT" similarly experiment groups for "N" and "O" was replaced with NT and OT. Similarly in figure 4 legend, indicated as Ctl/C, please clarify it.  This need to be fixed. 

Ans1: Thank you for your remind. We unified the acronym in this revision.

02)  Figure 3C 'y" axis indicated as normalized with CT = control..?,

Ans2: Thank you for your notice. CT was ICG signal intensity of control tumor. We renewed the label as “C” in this revision.

03)  Also in the same legend, "SEM. *p <0.05, # p <0.01, p <0.001" symbol for p <0.001= "&" need to include.     

Ans3: Thank you for your notice. We added the symbol “&” in this revision.

04) In Figure S1 legend, Western blot analysis NTCP (38kDa) and internal control a-tubulin (50kDa) in control and NTCP expressing HT-29. But in image it shows GAPDH as an internal control, which one is correct?   

Ans4: Thank you for your notice. The internal control for NTCP was a-tubulin (50kDa). It was a typo in the figure. We corrected in this revision.

Reviewer 2 Report

This paper shows the utility of the transport systems NTCP and OATP1B3, when expressed exogenously, in enabling them to uptake indocyanine green to facilitate detection of the expressing cells via fluorescence, and in the case of OAT1B3, MRI. This identifies a potentially useful experimental tool for detection of tumours and other cell types that can be made to express these transport molecules. However a key aspect in improving this paper would be comparison of cell sensitivity using this system with an existing infrared system.

One problem with the paper lies in the initial assessment of the transfected cells. In Figure S1a, phalloidin is used to denote the cell membrane, when in fact it is a marker for the actin cytoskeleton. In addition, when looking at these images, there is no clear evidence that the transporters are restricted to the cell membrane. In fact, particularly with the NTCP, a lot appears to be expressed in the middle of the cell, potentially in the nucleus. The authors should alter their description of these data to better reflect their findings, and consider using an alternative way of marking the cell membrane.

The data shown in Figure 1C are very hard to interpret – structures stained for DAPI seem to extensively overlap with the ICG staining and don’t look particularly like viable cells. Perhaps some control cells could be shown, along again with a cell membrane marker. But a more convincing demonstration of the cellular nature of these structures would improve the manuscript.

The in vivo data are reasonably convincing. However, to allow the reader to understand the utility of this system, it would be good to include a side by side comparison of a comparable infrared fluorescence system in the same cells.

Author Response

This paper shows the utility of the transport systems NTCP and OATP1B3, when expressed exogenously, in enabling them to uptake indocyanine green to facilitate detection of the expressing cells via fluorescence, and in the case of OAT1B3, MRI. This identifies a potentially useful experimental tool for detection of tumours and other cell types that can be made to express these transport molecules. However a key aspect in improving this paper would be comparison of cell sensitivity using this system with an existing infrared system.

Ans: Thank you for your comment. The title was changed to “ Use of indocyanine green (ICG), a medical near infrared dye, for enhanced fluorescent imaging – comparison of organic anion transporting polypeptide 1B3 (OATP1B3) and sodium-taurocholate cotransporting polypeptide (NTCP) reporter genes”

One problem with the paper lies in the initial assessment of the transfected cells. In Figure S1a, phalloidin is used to denote the cell membrane, when in fact it is a marker for the actin cytoskeleton. In addition, when looking at these images, there is no clear evidence that the transporters are restricted to the cell membrane. In fact, particularly with the NTCP, a lot appears to be expressed in the middle of the cell, potentially in the nucleus. The authors should alter their description of these data to better reflect their findings, and consider using an alternative way of marking the cell membrane.

Ans: Thank you for your comment. We correct our description to mention that NTCP was also expressed in the cytoplasm and the nucleus.

The data shown in Figure 1C are very hard to interpret – structures stained for DAPI seem to extensively overlap with the ICG staining and don’t look particularly like viable cells. Perhaps some control cells could be shown, along again with a cell membrane marker. But a more convincing demonstration of the cellular nature of these structures would improve the manuscript.

Ans: Thank you for your comment. Because there was high ICG intensity in NTCP-expressing HT-29, the saturated signal of ICG might overlap with the DAPI. In this revision, we changed the better data. This data will be easier to interpret.

The in vivo data are reasonably convincing. However, to allow the reader to understand the utility of this system, it would be good to include a side by side comparison of a comparable infrared fluorescence system in the same cells.

Ans: Thank you for your comment. We understand that your consideration. It is hard to inoculate control, NTCP, and OATP1B3 tumor into the same mice. Although we inoculated NTCP and OATP1B3 at different mice, we administrated ICG at the same dosage and the IVIS image was acquired at the same condition and the same day at each time point. Furthermore, we reorganized the figure make more comparable in this revision.

Round 2

Reviewer 2 Report

This paper is now acceptable